# Computational Insights into the Interplay of Mechanical Forces in Angiogenesis

**DOI:** 10.3390/biomedicines12051045

**Published:** 2024-05-09

**Authors:** Ana Guerra, Jorge Belinha, Christiane Salgado, Fernando Jorge Monteiro, Renato Natal Jorge

**Affiliations:** 1INEGI—Instituto de Ciência e Inovação em Engenharia Mecânica e Engenharia Industrial, 4200-465 Porto, Portugal; 2ISEP—Instituto Superior de Engenharia do Porto, Departamento de Engenharia Mecânica, Politécnico do Porto, Rua Dr. António Bernardino de Almeida, 431, 4249-015 Porto, Portugal; job@isep.ipp.pt; 3i3S—Instituto de Investigação e Inovação em Saúde, Universidade do Porto, 4200-135 Porto, Portugal; csalgado@ineb.up.pt (C.S.); fjmont@ineb.up.pt (F.J.M.); 4INEB—Instituto de Engenharia Biomédica, Universidade do Porto, 4200-135 Porto, Portugal; 5LAETA—Laboratório Associado de Energia, Transportes e Aeronáutica, Universidade do Porto, 4200-165 Porto, Portugal; rnatal@fe.up.pt; 6FEUP—Faculdade de Engenharia, Departamento de Engenharia Mecânica, Universidade do Porto, 4200-165 Porto, Portugal

**Keywords:** angiogenesis, computational modelling, compression loading, traction loading, radial point interpolation method

## Abstract

This study employs a meshless computational model to investigate the impacts of compression and traction on angiogenesis, exploring their effects on vascular endothelial growth factor (VEGF) diffusion and subsequent capillary network formation. Three distinct initial domain geometries were defined to simulate variations in endothelial cell sprouting and VEGF release. Compression and traction were applied, and the ensuing effects on VEGF diffusion coefficients were analysed. Compression promoted angiogenesis, increasing capillary network density. The reduction in the VEGF diffusion coefficient under compression altered VEGF concentration, impacting endothelial cell migration patterns. The findings were consistent across diverse simulation scenarios, demonstrating the robust influence of compression on angiogenesis. This computational study enhances our understanding of the intricate interplay between mechanical forces and angiogenesis. Compression emerges as an effective mediator of angiogenesis, influencing VEGF diffusion and vascular pattern. These insights may contribute to innovative therapeutic strategies for angiogenesis-related disorders, fostering tissue regeneration and addressing diseases where angiogenesis is crucial.

## 1. Introduction

Angiogenesis, the process of new blood vessel formation from pre-existing ones, plays a critical role in various physiological and pathological conditions [1]. It ensures an adequate blood supply to growing tissues, supporting their proper function and development. The interplay between mechanical and biological factors in promoting angiogenesis adds complexity to our understanding of this phenomenon.

In physiological scenarios, mechanical cues from the surrounding environment guide angiogenesis. For instance, during tissue development or repair, the mechanical forces generated by contracting muscles [2,3] or the need for increased blood flow [4,5] can trigger angiogenesis to ensure an adequate supply of nutrients and oxygen to the affected area. Conversely, in pathological conditions, the mechanical microenvironment of tissues can be altered, influencing angiogenesis. For example, tumour-associated mechanical forces, arising from increased interstitial pressure or altered extracellular matrix stiffness [6], can promote angiogenesis within the tumour microenvironment [7]. As presented, angiogenesis holds paramount importance in maintaining physiological homeostasis and contributing to the progression of several diseases [8]. The interplay between biological and mechanical factors in angiogenesis highlights the complexity of this process and opens ways for innovative therapeutic approaches that consider both aspects in the treatment of angiogenesis-related disorders.

Mechanical forces, such as compression and traction, exert profound effects on angiogenesis [9,10,11]. The application of mechanical forces can influence the release of angiogenic growth factors, like vascular endothelial growth factor (VEGF) [12], and modulate the behaviour of endothelial cells [13], which are essential for blood vessel formation. The mechanical forces of compression and traction are inherent to physiological environments, arising from factors such as muscle contractions, mechanical loading, and pressure from adjacent tissues [11,14,15]. These forces significantly impact the behaviour of blood vessels and the cells involved in angiogenesis, particularly endothelial cells.

Our study focuses on the computational simulation of these mechanical forces, using a model that incorporates the chemical diffusion of VEGF to regulate endothelial cell migration. In our previous studies, we developed a computational model that mimics in vivo angiogenesis. This model proves to be an effective tool for the analysis of the impact of VEGF diffusion on endothelial cell migration and the subsequent formation of capillary networks [16,17]. Computational simulation offers cost-effective and time-efficient approaches for studying complex phenomena such as angiogenesis [18,19]. Moreover, computational simulations enable the exploration of a wide range of scenarios and conditions, offering a high-throughput approach that allows researchers to efficiently explore numerous possibilities and to predict outcomes based on different scenarios, allowing them to guide experimental design and optimise resource utilisation. Additionally, in the context of angiogenesis [20,21,22,23], where the process is highly dynamic and influenced by numerous factors, computational simulation proves invaluable in deciphering the underlying mechanisms, exploring therapeutic interventions, and advancing our understanding of how mechanical forces, such as compression and traction, impact this vital biological phenomenon.

The impact of compression on angiogenesis has been explored in experimental studies [11,24,25,26,27]; however, the specific influence on the VEGF diffusion coefficient has remained largely unexamined. Our computational simulations, presented in this study, bridge this gap by providing a detailed analysis of the effect of compression and traction on capillary morphology and angiogenesis. Based on our simulations, it was demonstrated a notable promotion of angiogenesis through compression, resulting in enhanced capillary network density. Conversely, traction showed no significant effect on promoting angiogenesis. Nevertheless, our observations indicated that traction elevates the VEGF diffusion coefficient, fostering the migration of endothelial cells toward the region experiencing traction. These findings underscore the critical role of computational simulations in elucidating the mechanical aspects of angiogenesis, providing insights that complement experimental studies.

In summary, our study demonstrates the relationship between mechanical forces, specifically compression, and angiogenesis. By employing computational simulations, we contribute to the understanding of how these forces influence the VEGF diffusion coefficient and, consequently, capillary network formation. This knowledge not only advances our understanding of angiogenesis but also underscores the significance of integrating computational approaches to study the complexities of biological processes.

## 2. Materials and Methods

### 2.1. Numerical Method

In this study, the discretisation method used was the Radial Point Interpolation Method (RPIM) [28,29]. RPIM has previously proven its efficacy in modelling angiogenesis [16,30], and the comprehensive RPIM formulation can be found in the literature [31].

After identifying the problem domain and its essential natural boundary conditions, the domain can be discretised using a nodal set $N=\left\{n_{0},n_{1},\ldots ,\;n_{N}\right\}$. An integrated grid, either structured or unstructured, is subsequently established. Integration points within each grid cell are distributed according to a Gauss–Legendre integration scheme. Each integration point represents a partial volume of the problem complete volume, and this partial volume is defined by its associated weight, denoted as $\hat{\omega }$. The background integration mesh serves the purpose of numerically integrating the integro-differential equations that govern the physical phenomena being studied. Importantly, there are no predefined relationships between nodes and integration points, or even among nodes and integration points. RPIM stands out as a meshless method due to its lack of pre-established connectivity.

In meshless techniques, connectivity is determined by the influence domain concept. Every integration point $x_{I}$ searches radially for its n nearest nodes. According to the literature, it is advisable to maintain between 9 and 16 nodes within the influence domain of each integration point [31]. Connectivity between nodes is established through overlapping influence domains. Subsequently, shape functions are developed. RPIM employs the radial point interpolators method for shaping these functions [28,29]. For an integration point $x_{I}$, with n nodes within its influence domain, the interpolation function $u^{h}\left(x_{I}\right)$ can be defined as follows:(1)$$u^{h}\left(x_{I}\right)=\sum_{i=1}^{n}R_{i}\left(x_{I}\right)\cdot a_{i}\left(x_{I}\right)+\sum_{j=1}^{m}p_{j}\left(x_{I}\right)\cdot b_{j}(x_{I})=\left\{{R(x_{I})}^{T},{p(x_{I})}^{T}\right\}\left\{\begin{array}{c} a(x_{I})\\ b(x_{I}) \end{array}\right\}$$

The interpolation involves radial basis functions, denoted as $R\left(x_{I}\right)$, and a polynomial basis function, denoted as $p\left(x_{I}\right)$. The polynomial $p\left(x_{I}\right)$ is composed of m monomials, and the non-constant coefficients of each basis functions are represented by $a\left(x_{I}\right)$ and $b\left(x_{I}\right)$, respectively. These vectors can be expressed as follows:(2)$$R\left(x_{I}\right)={\left[R_{1}\left(x_{I}\right)R_{2}\left(x_{I}\right)\ldots R_{n}\left(x_{I}\right)\right]}^{T}$$
(3)$$P\left(x_{I}\right)={\left[p_{1}\left(x_{I}\right)p_{2}\left(x_{I}\right)\ldots p_{m}\left(x_{I}\right)\right]}^{T}$$
(4)$$a^{T}\left(x_{I}\right)=\left[a_{1}\left(x_{I}\right)a_{2}\left(x_{I}\right)\ldots a_{n}\left(x_{I}\right)\right]$$
(5)$$b^{T}(x_{I})=\left[b_{1}\left(x_{I}\right)\;b_{2}(x_{I})\ldots b_{m}(x_{I})\right]$$

The presented study employs the Multiquadrics Radial Basis Function (MQ-RBF) [28,31], defined as: $R_{iI}(x_{I})={\left(d_{Ii}^{2}+c^{2}\right)}^{p}$. The MQ-RBF shape parameters, c and p, have been previously tested and optimised for efficiency in prior research [31]. In this study, we adopt the recommended values: c=0.0001 and p=0.9999. Furthermore, $d_{Ii}$ characterises the Euclidean norm between the integration point $x_{I}={\left\{x_{I},y_{I}\right\}}^{T}$ and node $x_{i}={\left\{x_{i},y_{i}\right\}}^{T}$, $d_{Ii}=\sqrt{{\left(x_{i}-x_{I}\right)}^{2}+{\left(y_{i}-y_{I}\right)}^{2}}$. Concerning the polynomial basis function, $p\left(x_{I}\right)$, this work uses its linear version:(6)$${p(x)}^{T}=\left[1,x,y\right]$$Consequently, m=3. An additional equation system must be included to obtain an unique solution [30]:(7)$$\sum_{\;i=1}^{n}p_{j}(x_{i})a_{i}(x_{i})=0$$

Applying Equations (1) and (7) for every node within the influence domain of $x_{I}$, the following system of equations is achieved:(8)$$\left\{\begin{array}{c} u_{s}\\ 0 \end{array}\right\}=\left[\begin{array}{cc} R & P\\ P^{T} & 0 \end{array}\right]\left\{\begin{array}{c} a\\ b \end{array}\right\}=G\left\{\begin{array}{c} a\\ b \end{array}\right\}$$

The radial moment matrix is defined by R, the polynomial moment matrix by P, and the vector of the nodal values by $u_{s}:$(9)$$R=\left[\begin{array}{ccc} R(r_{11}) & R(r_{21}) & \begin{array}{cc} \ldots & R(r_{1n}) \end{array}\\ R(r_{21}) & R(r_{22}) & \begin{array}{cc} \ldots & R(r_{2n}) \end{array}\\ \begin{array}{c} \vdots \\ R(r_{n1}) \end{array} & \begin{array}{c} \vdots \\ R(r_{n2}) \end{array} & \begin{array}{c} \;\vdots \\ \begin{array}{cc} \ldots & R(r_{nn}) \end{array} \end{array} \end{array}\right]$$
(10)$$P=\left[\begin{array}{ccc} 1 & x_{1} & y_{1}\\ 1 & x_{2} & y_{2}\\ \vdots & \vdots & \vdots \\ 1 & x_{n} & y_{n} \end{array}\right]$$
(11)$$u_{s}={\left\{u_{1},u_{2},\ldots \;u_{n}\right\}}^{T}$$

Thus, the non-constant coefficients can be derived as follows:(12)$$\left\{\begin{array}{c} a\\ b \end{array}\right\}=G^{-1}\left\{\begin{array}{c} u\\ 0 \end{array}\right\}$$

Substituting the result from Equation (12) into Equation (1):(13)$$u^{h}\left(x_{I}\right)=\left\{R^{T}(x_{I})p^{T}(x_{I})\right\}G^{-1}\left\{\begin{array}{c} u\\ 0 \end{array}\right\}=\left\{\varphi \left(x_{I}\right),\Psi \left(x_{I}\right)\right\}\left\{\begin{array}{c} u\\ 0 \end{array}\right\}$$
where $\varphi \left(x_{I}\right)\;$is the RPI shape function vector that is defined as follows:(14)$$\varphi \left(x_{I}\right)=\left\{\varphi_{1}\left(x_{I}\right)\;\varphi_{2}\left(x_{I}\right)\;\ldots \;\varphi_{n}\left(x_{I}\right)\right\}$$

And, $\Psi \left(x_{I}\right)\;$the residual vector defined as follows:(15)$$\Psi \left(x_{I}\right)=\left\{\Psi_{1}\left(x_{I}\right)\;\Psi_{2}\left(x_{I}\right)\;\ldots \;\Psi_{m}\left(x_{I}\right)\right\}$$

The spatial partial derivatives of $\varphi \left(x_{I}\right)$ are readily acquired, as detailed in [31]. Moreover, $\varphi \left(x_{I}\right)\;$features the Kronecker delta property$\;\delta_{ij}$, allowing to impose the essential boundary conditions within the stiffness matrix, similar to other interpolation techniques like the finite element method [32]. Additionally, the compact support of $\varphi \left(x_{I}\right)$ facilitates the creation of banded stiffness matrices [31], thereby reducing computational costs.

### 2.2. Reaction–Diffusion System

Modelling endothelial cell migration during angiogenesis, as described in previous studies [16,30], entailed accounting for the chemical diffusion of VEGF within a homogeneous medium. Additionally, numerical simulation of VEGF diffusion as a field problem is feasible [33]. Utilising the meshless formulation, the discrete equation system for the variable field being analysed can be established through the general formulation of the Helmholtz equation (Equation (16).
(16)$$D_{x}\frac{\partial^{2}\phi }{\partial x^{2}}+D_{y}\frac{\partial^{2}\phi }{\partial y^{2}}-g\phi +Q=0$$
where ϕ is the field variable that corresponds to the VEGF concentration, $D_{x}$ and $D_{y}$ represent the VEGF diffusion coefficient along dimensions x and y, respectively, g represents the matrix of chemical infusibility (in this work g is neglected, g=0) and Q is the VEGF release rate. Using the weighted residual approach, the following meshless system equations can be formulated:(17)$$\left[K_{D}+K_{g}\right]\Phi -f_{q}=0$$
where $K_{D}$, after the manipulation of Equation (16), can be represented as
(18)$$K_{D}=\int_{A}B_{G}^{T}DB_{G}dA$$
being $B_{G}$ and D, respectively, defined as
(19)$$B_{G}=\left[\begin{array}{c} \frac{\partial \varphi }{\partial x}\\ \frac{\partial \varphi }{\partial y} \end{array}\right]=\left[\begin{array}{ccc} \frac{\partial \varphi_{1}}{\partial x} & \frac{\partial \varphi_{2}}{\partial x} & \ldots \;\frac{\partial \varphi_{n}}{\partial x}\\ \frac{\partial \varphi_{1}}{\partial y} & \frac{\partial \varphi_{2}}{\partial y} & \ldots \;\frac{\partial \varphi_{n}}{\partial y} \end{array}\right]$$
(20)$$D=\left[\begin{array}{cc} D_{x} & 0\\ 0 & D_{y} \end{array}\right]$$

Additionally, $K_{g}$ can be defined as
(21)$$K_{g}=\int_{A}g{\left\{\begin{array}{cccc} \varphi_{1} & \varphi_{2} & \ldots & \varphi_{n} \end{array}\right\}}^{T}\left\{\begin{array}{cccc} \varphi_{1} & \varphi_{2} & \ldots & \varphi_{n} \end{array}\right\}dA$$
and $f_{q}$, the discrete VEGF release rate vector, is defined as
(22)$$f_{q}=\int_{A}Q{\left\{\begin{array}{cccc} \varphi_{1} & \varphi_{2} & \ldots & \varphi_{n} \end{array}\right\}}^{T}dA$$

Using this methodology, and upon solving Equation (17), the final VEGF concentration in the medium, denoted as Φ, can be determined. This VEGF concentration governs endothelial cell migration and angiogenesis.

The resulting equation system can be represented as follows:(23)$$K\Phi -f_{q}=0$$
being K = $K_{D}+K_{g}$.

Due to the delta Kronecker property inherent in RPIM shape functions, the direct imposition method is employed to impose periodic boundary conditions. Consequently, by imposing identical VEGF concentrations at node $n_{i}$ (associated with the left essential boundary) and node $n_{j}$ (associated with the right essential boundary), modifications are applied to the stiffness matrix in the ith degrees of freedom.
(24)$$K_{im}=0\;⟶m=1,\ldots ,N\;K_{ii}=1\;\wedge K_{ij}=-1$$
and in jth degree of freedom as
(25)$$K_{jm}=0\;⟶m=1,\ldots ,N\;K_{jj}=1\;\wedge K_{ji}=-1$$
where N represents the total number of degrees of freedom (in this case, the only variable is the VEGF concentration, n is equal to the total number of nodes discretising the problem domain).

### 2.3. Mechanical Loading Application

To simulate the domain stress–strain interactions, the elasticity theory was included into the algorithm. Considering linear elastostatic relations, the following equilibrium equations are considered at domain Ω:(26)$$\left\{\begin{array}{c} \nabla \Lambda +b=0\;\mathrm{in}\;Ω\;\\ \Lambda \;n=\bar{t}\;\mathrm{on}\;\Gamma_{t}\\ u=\bar{u}\;\mathrm{on}\;\Gamma_{u} \end{array}\right.$$
where ∇ represents the gradient operator, u the displacement field, Λ the Cauchy stress tensor, b the body force per unit volume, and Γ the contour containing the essential and the natural boundaries, $\Gamma_{u}$ and $\Gamma_{t}$, respectively. Note that $\Gamma \in Ω:\Gamma_{u}\cup \Gamma_{t}=\Gamma ⋀\Gamma_{u}\cap \Gamma_{t}=\emptyset$ and $\bar{u}$ represents the prescribed displacement on $\;\Gamma_{u}$, $\bar{t}$ the traction on $\;\Gamma_{t}$ and n the unit outward normal to Γ.

Applying the weak form of Galerkin, the following equation is obtained:(27)$$\delta L=\int_{Ω}\delta \varepsilon^{T}\sigma dΩ-\int_{Ω}\delta u^{T}bdΩ-\int_{\Gamma_{t}}\delta u^{T}\bar{t}\;d\Gamma_{t}=0$$
where ε is the strain vector and σ is the stress vector, both in Voigt notation. The strain vector, ε**,** and the stress vector, σ, using the Hooke’s law, can be defined, respectively, as follows:(28)$$\varepsilon =Lu=\left[\begin{array}{cc} \frac{\partial }{\partial x} & 0\\ 0 & \frac{\partial }{\partial y}\\ \frac{\partial }{\partial y} & \frac{\partial }{\partial x} \end{array}\right]\cdot \left\{\begin{array}{c} u\\ v \end{array}\right\}$$
(29)$$\sigma =c\varepsilon =cLu={\left[\begin{array}{ccc} \frac{1-\upsilon^{2}}{E} & -\frac{\upsilon +\upsilon^{2}}{E} & 0\\ -\frac{\upsilon +\upsilon^{2}}{E} & \frac{1-\upsilon^{2}}{E} & 0\\ 0 & 0 & \frac{1}{G} \end{array}\right]}^{-1}\cdot \left(\left[\begin{array}{cc} \frac{\partial }{\partial x} & 0\\ 0 & \frac{\partial }{\partial y}\\ \frac{\partial }{\partial y} & \frac{\partial }{\partial x} \end{array}\right]\cdot \left\{\begin{array}{c} u\\ v \end{array}\right\}\right)$$
where c is the material constitutive matrix, E is the Young’s modulus, υ is the material Poisson’s coefficient and G is the elastic shear modulus (G=E/(2+2υ)). Upon manipulation [31], Equation (27) can be written as
(30)$$Ku=f_{b}+f_{t}$$
being,
(31)$$K=\int_{Ω}B^{T}\cdot c\cdot BdΩ=\sum_{i=1}^{n_{Q}}B_{I}^{T}\cdot c_{I}\cdot B_{I}\cdot \hat{\omega_{I}}$$
(32)$$f_{b}=\int_{Ω}H^{T}\cdot bdΩ=\sum_{i=1}^{n_{Q}}H_{I}^{T}\cdot b_{I}\cdot \hat{\omega_{I}}$$
and,
(33)$$f_{t}=\int_{Ω}H^{T}\cdot \bar{t}dΩ=\sum_{i=1}^{n_{q}}H_{I}^{T}\cdot {\bar{t}}_{I}\cdot \hat{\omega_{I}}$$

Matrices $B_{I}$ and $H_{I}$ are defined as
(34)$$B_{I}=\left[\begin{array}{ccccccc} \frac{\partial \varphi_{1}\left(x_{I}\right)}{\partial x} & 0 & \frac{\partial \varphi_{2}\left(x_{I}\right)}{\partial x} & 0 & \cdots & \frac{\partial \varphi_{n}\left(x_{I}\right)}{\partial x} & 0\\ 0 & \frac{\partial \varphi_{1}\left(x_{I}\right)}{\partial y} & 0 & \frac{\partial \varphi_{2}\left(x_{I}\right)}{\partial y} & \cdots & 0 & \frac{\partial \varphi_{n}\left(x_{I}\right)}{\partial y}\\ \frac{\partial \varphi_{1}\left(x_{I}\right)}{\partial y} & \frac{\partial \varphi_{1}\left(x_{I}\right)}{\partial x} & \frac{\partial \varphi_{2}\left(x_{I}\right)}{\partial y} & \frac{\partial \varphi_{2}\left(x_{I}\right)}{\partial x} & \cdots & \frac{\partial \varphi_{n}\left(x_{I}\right)}{\partial y} & \frac{\partial \varphi_{n}\left(x_{I}\right)}{\partial x} \end{array}\right]$$
and,
(35)$$H_{I}=\left[\begin{array}{ccccccc} \varphi_{1}\left(x_{I}\right) & 0 & \varphi_{2}\left(x_{I}\right) & 0 & \cdots & \varphi_{n}\left(x_{I}\right) & 0\\ 0 & \varphi_{1}\left(x_{I}\right) & 0 & \varphi_{2}\left(x_{I}\right) & \cdots & 0 & \varphi_{n}\left(x_{I}\right) \end{array}\right]$$
$n_{Q}$ is the number of integration points discretising the problem domain, $n_{q}$ is the number of integration points at the traction boundary and $c_{I}$ is the constitutive matrix of integration point $x_{I}$.

### 2.4. Model Parameters

#### 2.4.1. Domain Geometry 

The simulation of capillary network growth was conducted within 5 × 5 mm^2^ square domain, discretised with 2601 nodes to create a regular nodal mesh. In this study, three different initial domain geometries were defined, as illustrated in Figure 1, aligning with approaches from previous works [16,30,34]. In the performed simulations, the location of the capillary lumen, the endothelial cell monolayer and the VEGF release region change in each model under analysis.

#### 2.4.2. Boundary Conditions

Concerning the essential boundary conditions, a VEGF basal concentration of 2.35 × 10^−13^ g mm^−3^ was applied in the capillary lumen and a concentration of 6.43 × 10^−13^ g mm^−3^ was applied in the VEGF release region [35]. Regarding the natural boundary conditions, a VEGF flux from the VEGF release region was applied. The VEGF chemical diffusion coefficient used was 1.16 × 10^−6^ mm^2^ s^−1^ [36]. With this methodology, the VEGF gradient concentration that rules the endothelial cell migration during angiogenesis was obtained (Figure 2). This gradient goes from dark blue to yellow. The dark blue zone is located near the endothelial cell monolayer and has a VEGF concentration of around 2.35 × 10^−13^ g mm^−3^. The yellow zone is located near the VEGF release region and has the highest VEGF of around 6.43 × 10^−13^ g mm^−3^.

In this study, the impact of applying mechanical loading on angiogenesis was investigated, specifically examining its influence on the VEGF diffusion gradient. The details of this effect are expounded below.

A hydrostatic pressure of $p_{H}=0.0067$ MPa (50 mmHg) [37] was applied to a specific zone within the domain, which in this work corresponds to the central circle in Figure 3. Considering a positive hydrostatic pressure will enable the analysis of the influence of the VEGF diffusion gradient under tensile stress applied to the medium, while assuming negative hydrostatic pressure will allow the study of the effect of compressive stress.

To incorporate the effect of hydrostatic pressure in the medium, a mathematical model was assumed. In this work, it was hypothesised that the VEGF diffusion coefficient in the medium is influenced by the pressure within its domain. Consequently, a medium subjected to compression stress is anticipated to exhibit a lower diffusion coefficient compared to the basal value, as compression results in a more compact medium. Conversely, a tensile stress is expected to elevate the diffusion coefficient above the basal level. Additionally, it was assumed that the pressure decreases in a parabolic manner from the centre of the circle to its periphery:(36)$$f=\alpha \cdot \;(1-{d_{ii}}^{2}/{r_{c}}^{2}\;)$$
where $r_{c}$ is the centre of the circle where hydrostatic pressure is applied and $d_{ii}$ represents the distance to the centre of the circle, and it is defined as$\;d_{IC}=\sqrt{{\left(x_{I}-x_{c}\right)}^{2}+{\left(y_{I}-y_{c}\right)}^{2}},\;$ with $x_{c}$ and $y_{c}$ being the circle centre coordinates and $x_{I}\;$and $y_{I}\;$the coordinates of the interest point. If $d_{ii}>r_{c},$ then f=0.

Regarding the alpha parameter, it was defined as follows:(37)$$\alpha =\frac{\bar{p_{H}}\;}{p_{H}}\;$$
where $\bar{p_{H}}\;$is the hydrostatic pressure calculated by the mathematical code in the interest point. In this study, different alpha values were analysed (0.5, 1.0 and 1.5).

Then, the VEGF diffusion coefficient within the circle was modified accordingly to the following expression:(38)$$D_{V{EGF}^{H}}=D_{VEGF}+(D_{VEGF}\cdot \;f)$$
with $D_{VEGF}$ being the basal VEGF diffusion coefficient.

### 2.5. Angiogenesis Modulation

In our model, angiogenesis initiation involves identifying the initial sprouting cells within a designated sub-domain, using Cartesian coordinates. The initial number of sprouting cells and its location vary in each simulation. Following this, an iterative loop is initiated. At each time step, tip cells are sequentially marked to facilitate capillary network growth. Each time step corresponds to the duration for capillaries to migrate a small cluster of endothelial cells. The direction of endothelial cell movement is determined by vector u, which follows the gradient vector from the VEGF concentration field toward the VEGF source. Consequently, the gradient of the VEGF concentration field (n=∇Φ) is obtained, with n being oriented towards the VEGF source, opposite to its support cell.

Accordingly, $n_{mod}=\left[\begin{array}{cc} cos\theta & sin\theta \\ -sin\theta & cos\theta \end{array}\right]n$. Then, the unit vector of $n_{mod}$ is calculated as follows: $u=n_{mod}/||n_{mod}||$. Hence, utilising the directional vector u, the average cell-to-cell distance is factored in to determine the new position of the tip cell. This is achieved by adding the old tip cell position to the product of vector u and the average cell-to-cell distance. If a node occupies the new tip cell position, it transforms into a tip cell. Conversely, if no nodes are present at the new position, a new node is introduced into the domain, subsequently becoming a tip cell. During each iteration, the VEGF concentration is computed, and the tip cell’s new position is updated accordingly. Additionally, the VEGF concentration at the boundary domain is enforced. The iterative process stops when the tip cell reaches the domain boundary, reaches the VEGF release region, or approaches another endothelial cell closely (half of the internodal distance).

The methodology for implementing the branching process is outlined in a previous work [16]. Subsequently, capillary network observed in an in vivo angiogenesis assay was analysed to establish a phenomenological law linking the distance between consecutive branches and the capillary order. Capillary order was categorised based on capillary calibre, with the first order representing capillaries of larger calibre and the third order representing those of smaller calibre. Accordingly, a new branch occurs if the distance to the previous branch is greater than the calculated distance for the capillary order considered.

### 2.6. Compression and Traction Effect on Angiogenesis Implementation

Most experimental studies indicate that compression has a more predominant role than traction in promoting angiogenesis [11,24,25,37]. Thus, in this study, the effect of compression on branching is analysed and the obtained numerical results were compared with experimental data.

To achieve this, a modification in the phenomenological law governing branching occurrence was implemented into the algorithm. Accordingly, when the tip cell enters the zone where compression is applied, the distance required for a new branching, as calculated by the previously described phenomenological law, is updated and reduced by half. Utilising this approach within the computational model, we simulated the impact of compressive loading on angiogenesis. To assess this impact, we the total number and length of vessels were calculated from all traced vessels. Subsequently, the numerical findings were juxtaposed with experimental findings sourced from the literature.

Given the scarcity of experimental studies evaluating the effect of traction on angiogenesis, particularly on endothelial cell migration, the occurrence of branching remained unaltered and was simulated as in previous studies [16,17]. For the same reason, the comparison of numerical results obtained for traction with experimental data was compromised.

## 3. Results

### 3.1. Effect of Compression and Traction in VEGF Diffusion Coefficient

Compression and traction are two crucial mechanical forces that play a significant role in angiogenesis. Compression, as a mechanical force, is primarily exerted on tissues and blood vessels in response to external factors. This force can arise from various sources, such as muscle contractions, mechanical loading, or the pressure applied by adjacent tissues. When blood vessels experience compression, it initiates a series of cellular responses resulting in the activation of endothelial cells.

In our model, endothelial cell migration is regulated by the chemical diffusion of VEGF. Accordingly, the influence of compression on the VEGF diffusion coefficient was analysed and, consequently, in the VEGF gradient. Figure 4 demonstrates that compression leads to a reduction in the VEGF diffusion coefficient. Notably, this reduction is directly proportional to the increment in the alpha parameter value. This reduction, in turn, impacts the VEGF concentration within the region where compression is applied. As a consequence of this change, endothelial cell migration is directed away from the area subjected to compression.

Several simulations were conducted with distinct values for the alpha parameter to fine-tune its optimal setting. Notably, when alpha was set to 1.5, it was observed with a degree of instability in the results. Therefore, in our subsequent simulations, an alpha value of 1.0 was adopted.

Traction is generated within cells and tissues due to the mechanical forces applied to them. It is a result of the interaction between cells and the extracellular matrix and can induce endothelial cell migration and alignment. Accordingly, analysing the effect of traction in cell behaviour and in angiogenesis is a difficult task and there is a lack of experimental results in this matter. Nevertheless, in this study, it was assumed that traction would have an opposite effect on the VEGF diffusion coefficient compared to compression. Employing this hypothesis, we conducted a numerical analysis to explore the impact of traction on the VEGF diffusion coefficient and, consequently, its influence on endothelial cell migration. Figure 5 reveals that traction leads to an increase in the VEGF diffusion coefficient. As in the previously case, this increase is directly proportional to the increment in the alpha parameter value. This increase, in turn, affects the VEGF concentration within the region where traction is applied. As a consequence of this change, endothelial cells migrate closer to the region where traction is being applied.

### 3.2. Effect of Compression and Traction in Angiogenesis

The impact of compression on angiogenesis has been explored in some experimental studies [11,24,25,37]. However, the impact of traction on angiogenesis has not been previously investigated, and the precise influence of both compression and traction on the VEGF diffusion coefficient remains unexplored. Therefore, in this study, four distinct simulations were performed aiming to study the impact of compression and traction on capillary morphology and angiogenesis. These simulations involved variations in the parent vessel location and the region of VEGF release. Furthermore, the initial number of sprouting cells differed across these simulations.

Upon examination of Figure 6, it becomes evident that compression indeed promotes angiogenesis. In the region subjected to compression, there is a noticeable augmentation in the capillary network density. To substantiate this observation, we quantified both the total vessel length and the number of branches (Figure 7). Traction does not seem to have an effect on promoting angiogenesis. Analysing the capillary pattern obtained in the simulations (Figure 6), there does not appear to be an increase in capillary density. However, it seems that the migration of endothelial cells toward the region under traction is promoted, and this effect is more pronounced in Example 1.

Upon analysing Figure 7, it becomes evident that compression significantly enhances angiogenesis. In Example 1, compression resulted in a 68% increase in vessel length and a 70% increase in the number of branches. Similarly, in Example 2, compression led to a 15% increase in vessel length and a 31% increase in the number of branches. In Example 3, compression increased vessel length by 19% and the number of branches by 45%. Finally, in Example 4, compression brought about an 18% increase in vessel length and a 35% increase in the number of branches. The quantitative results obtained for traction are similar to those obtained for the situation without mechanical stimulation.

## 4. Discussion

Endothelial cells and blood vessels inhabit environments characterised by continual mechanical activity. Previous studies have demonstrated the significant influence of factors such as fluid shear stress, stretching, compression, and hydrostatic pressure on various cellular processes, including signal transduction, cytoskeletal organisation, gene expression, as well as in endothelial cell migration, proliferation, and extracellular matrix remodelling [37,38,39,40]. Understanding the response of endothelial cells to mechanical triggers is essential for enhancing tissue vascularisation and advancing the prospects of wound repair and tissue engineering.

In this study, one of the goals was to explore the impact of mechanical forces, specifically compression and traction, on angiogenesis using a meshless computational method. To achieve this, three distinct initial domain geometries were defined, each varying the initial location for sprouting endothelial cells and VEGF release region in alignment with established methodologies from prior works [16,30,34]. Then, based on experimental findings [37], a hydrostatic pressure of 0.0067 MPa (50 mmHg) was applied to a specific zone within the domain. Subsequently, we analysed the effects of applying both tensile and compressive loads on the VEGF diffusion coefficient, and its consequent influence on the final VEGF diffusion gradient (Figure 4 and Figure 5).

The presented documented numerical findings showed that compression results in a proportional reduction in the VEGF diffusion coefficient, impacting VEGF concentration in the compressed region. Consequently, endothelial cell migration shifts away from the compressed area. The findings align with experimental results demonstrating that compression decreases the diffusion coefficient. This alignment is corroborated by studies on the rate of self-diffusion in zinc single crystals [41] and on the effects of compression on macromolecular diffusion in articular cartilage, which indicate a notable reduction in diffusivity under high strains in the surface zone [42]. Additionally, there is evidence of a significant decrease in glucose diffusivity with increasing static compressive strain [43]. Moreover, in an ex vivo culture model of rabbit vertebral endplate, the application of constant compressive load led to a gradual reduction in the number of vascular buds and a significant decrease in both VEGFA and VEGFR2 protein concentrations [44].

Analysing traction, generated by mechanical forces within cells and tissues, presents a challenge due to a scarcity of experimental results. While existing numerical studies have concluded that cell-generated traction forces influence the migration, proliferation, and differentiation of various cell phenotypes, including endothelial cells and pericytes involved in angiogenesis [45,46,47], none have explored the impact of traction on tissue and its subsequent effects on endothelial cell behaviour and blood vessel formation. However, our numerical analysis reveals that traction increases the VEGF diffusion coefficient, influencing VEGF concentration and prompting endothelial cells to migrate towards the region experiencing traction. This observation is intriguing and should be further explored in future studies, given its potential to affect angiogenesis in scenarios like wound healing or the vascularisation of scaffolds. Considering the number of vessels and branches, we did not observe in our simulations that traction promotes angiogenesis. However, this may have occurred because due to the scarcity of experimental studies, angiogenesis was simulated as in previous studies. We did not have robust data to modify our computational model in this respect.

This study focused on analysing the effect of compression on branching and its subsequent impact on the capillary network. To achieve this, a modification was introduced to the phenomenological law governing branching occurrence described in previous works [16,17]. This modification involved updating and halving the distance required for a new branching when the tip cell entered the zone experiencing compression. Employing this approach within the proposed computational model allowed the simulation of the impact of compressive loading on growth of blood vessels from the existing vasculature. While previous experimental studies have explored the impact of compression on angiogenesis [11,24,25,37], this study provides a distinctive contribution by specifically examining the influence of compression on the VEGF diffusion coefficient. Through four distinct simulations that varied in parent vessel location, VEGF release region, and initial sprouting cell numbers, the obtained results (Figure 6) demonstrated that compression significantly promotes angiogenesis. The region subjected to compression exhibited an increase in capillary network density. This observation was further corroborated through quantitative analyses of total vessel length and the number of branches (Figure 7). For instance, in Example 1, compression led to a 68% increase in vessel length and a 70% increase in the number of branches. Similar trends were observed in Examples 2, 3, and 4, confirming the consistent angiogenic promotion facilitated by compression in diverse simulation scenarios. The present documented findings are consistent with the experimental findings reported by Ruehle et al. [11]. In their study, gels containing microvascular fragments were subjected to compressive indentation using platens with a smaller diameter than that of the gel. Specifically, for a 10% strain, they observed a 52% increase in microvascular network length and a 76% increase in the number of branch points in microvascular fragments under continuous loading over the final 5 days of culture. Additionally, Yoshino et al. [37] found that exposure to hydrostatic pressure (+50 mmHg) increased the formation of tube-like structures by endothelial cells by 32% and enhanced the number of branch points by 147%.

Despite these valuable insights, this study has its limitations. The simplified representation of mechanical loading, absence of blood flow considerations, and the static nature of the mechanical environment pose challenges in fully capturing the dynamic and heterogeneous conditions observed in vivo. The need for experimental validation, especially in the analysis of traction forces in tissue, and the potential impact of varying loading conditions over time demand further investigation. The findings of this study contribute to the growing knowledge on the mechanical aspects of angiogenesis, offering a possibility for future research. Refinements in computational models, incorporation of blood flow dynamics and validation of numerical results with experimental studies are crucial steps to enhance the reliability and applicability of our results. The observed effects of compression on angiogenesis have implications for fields such as tissue engineering, wound healing, and regenerative medicine, where understanding the mechanical regulation of vascular development is pivotal.

## 5. Conclusions

In summary, this study provides a comprehensive computational exploration of interplay between mechanical forces, specifically compression and traction, and angiogenesis. We observed that compression significantly promotes angiogenesis, as evidenced by increased capillary network density and validated through quantitative analyses of vessel length and branching number. The reduction in the VEGF diffusion coefficient under compression, influencing VEGF concentration and altering endothelial cell migration patterns, underscores the mechanical regulation of angiogenesis. The approach outlined in this work offers valuable insights for devising innovative therapeutic methods to address angiogenesis-related disorders. By unravelling the mechanical cues governing angiogenesis, researchers and clinicians can unlock innovative approaches to enhance blood vessel formation and improve the overall health and well-being of individuals.

## Figures and Tables

**Figure 1 biomedicines-12-01045-f001:**
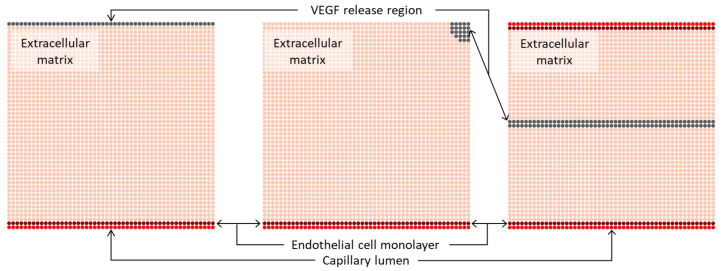
Initial domain geometries used in this study. The location of the capillary lumen, the endothelial cell monolayer and the VEGF release region change in the performed simulations. Label in red represents the capillary lumen, label in dark red represents the endothelial cell monolayer, label in light pink represents the extracellular matrix, and label in grey represents the VEGF release region.

**Figure 2 biomedicines-12-01045-f002:**
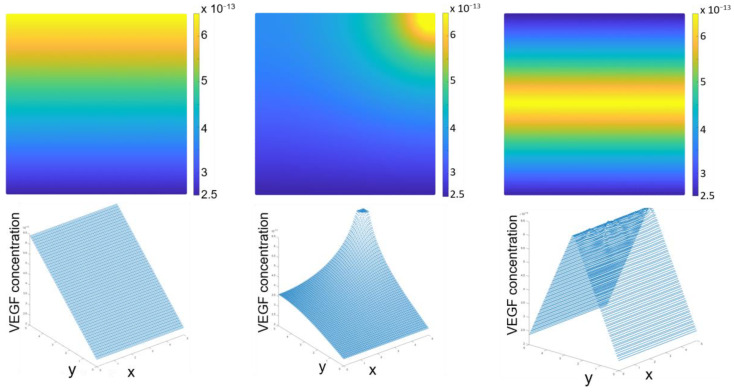
VEGF diffusion gradient. The dark blue zone is located near the endothelial cell monolayer and has the basal VEGF concentration. The yellow zone is located near the VEGF release region and has the highest VEGF concentration. (VEGF concentration in g mm^−3^. The scale bar used is linear).

**Figure 3 biomedicines-12-01045-f003:**
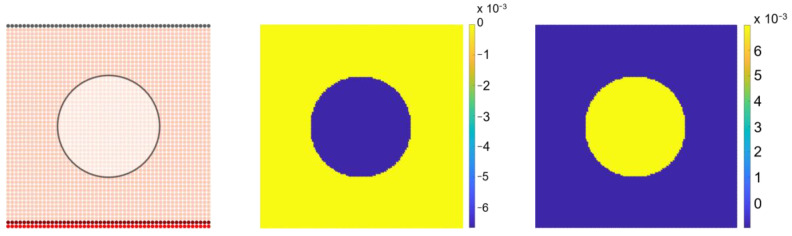
Mechanical loading application in a circular region in the middle of the domain (**Left**, color labels are related to Figure 1) and the respectively obtained compression (**middle**) and traction (**right**) colourmaps (Scale bar in MPa: ×10^−3^. The scale bar used is linear).

**Figure 4 biomedicines-12-01045-f004:**
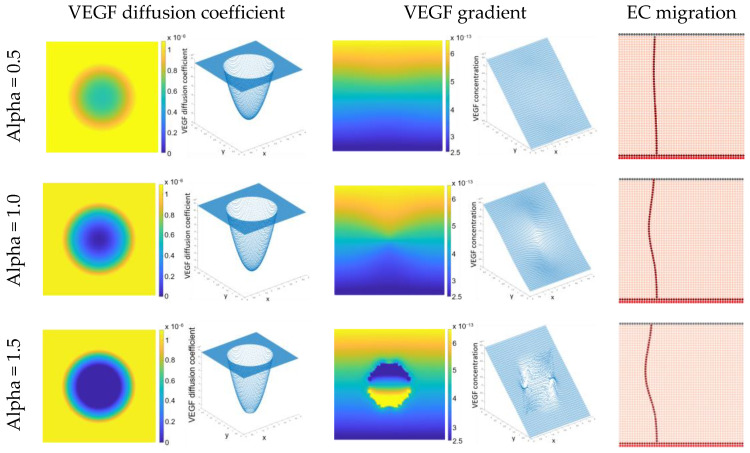
Effect of compression in VEGF diffusion coefficient (in mm^2^ s^−1^) and consequently, in VEGF gradient (in g mm^−3^) and endothelial cell migration (color labels are related to Figure 1).

**Figure 5 biomedicines-12-01045-f005:**
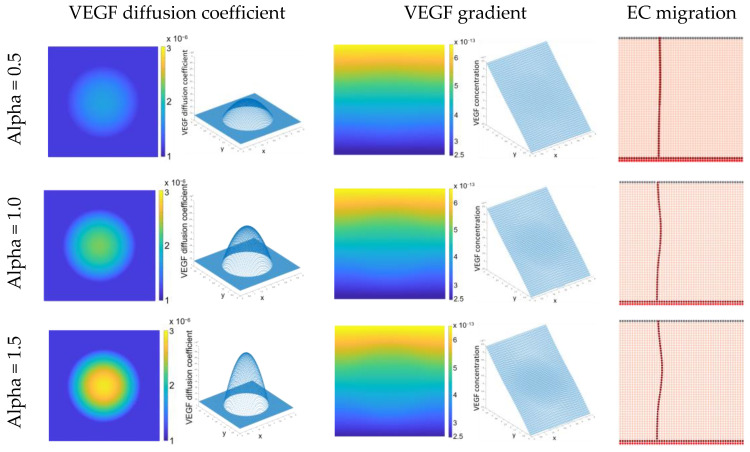
Effect of traction in VEGF diffusion coefficient (in mm^2^ s^−1^) and consequently, in VEGF gradient (in g mm^−3^) and endothelial cell migration (color labels are related to Figure 1).

**Figure 6 biomedicines-12-01045-f006:**
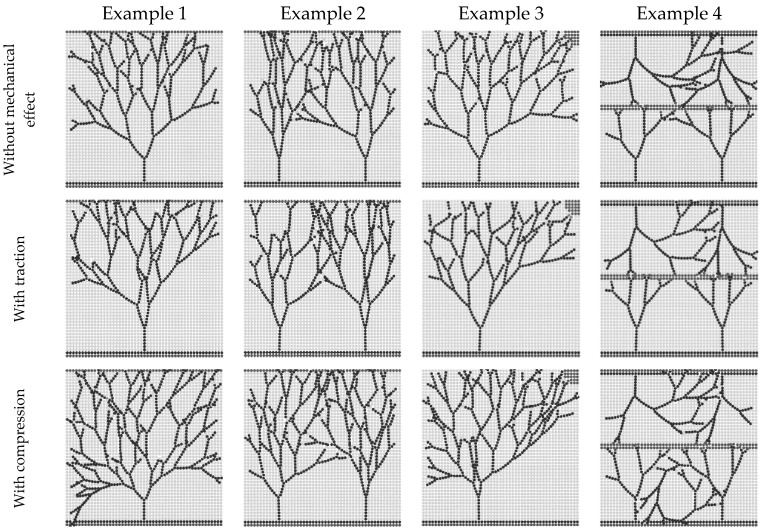
Final simulation results obtained for angiogenesis, using a regular nodal discretisation mesh. In Examples 1 and 2, the parent vessel is located at the lower domain boundary, while the VEGF release region is situated at the upper boundary. These examples feature one and two initial sprouting cells, respectively. In Example 3, the parent vessel is located at the lower domain boundary, the VEGF release region is positioned in the upper right corner, and there is a single initial sprouting cell. In Example 4, two parent vessels are found at the lower and upper domain boundaries, the VEGF release region is situated in the middle domain, and four initial sprouting cells are presented. The images are sized at 5 mm × 5 mm.

**Figure 7 biomedicines-12-01045-f007:**
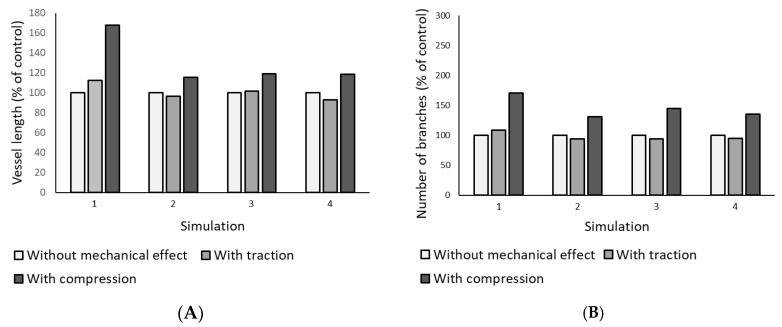
Quantification of vessel length (**A**) and number of branches (**B**) for each performed simulation domain. The results are presented as a percentage of the control.

## Data Availability

Data are contained within the article.

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
