# Peer review of "Computational Insights into the Interplay of Mechanical Forces in Angiogenesis"

_biomedicines, 2024, doi:10.3390/biomedicines12051045_

Round 1
Reviewer 1 Report
Comments and Suggestions for Authors
This manuscript presented a numerical model for studying the influence of mechanical forces (compression and traction) on the angiogenesis. In the model they designed, the VEGF diffusion coefficient changes under the external forces, and thus affects the process of angiogenesis. The topic and the results of the investigation should be interesting for the readers of the journal. However, in mu opinion, this manuscript should be significantly improved before it can be accepted for publishing.
1. The authors didn’t clearly describe how the mechanical forces affects the VEGF diffusion coefficient. Is there any physiological evidence quantitatively supporting this connection?
2. In Fig. 7, the authors compared their results with the experiments available, which indicated that they are in the same order. However, for the main topic of the study, I didn’t find the experimental results to support their results on the influence of compression and traction on the angiogenesis in the figure.
3. It seems that there are some mathematical symbols are misused, which leads to some confusion. For example, is the u_s in eq 8 same as u in eq 12 and 13? There is another u in eq 26. Are they have same meaning? Please check your writing carefully.
4. Many words in the figures are too small to be distinguished and read clearly.
5. The writing of the manuscript should be improved. Many sentences in the article are vague and not scientifically precise enough.
Comments on the Quality of English LanguageThis manuscript presented a numerical model for studying the influence of mechanical forces (compression and traction) on the angiogenesis. In the model they designed, the VEGF diffusion coefficient changes under the external forces, and thus affects the process of angiogenesis. The topic and the results of the investigation should be interesting for the readers of the journal. However, in mu opinion, this manuscript should be significantly improved before it can be accepted for publishing.
1. The authors didn’t clearly describe how the mechanical forces affects the VEGF diffusion coefficient. Is there any physiological evidence quantitatively supporting this connection?
2. In Fig. 7, the authors compared their results with the experiments available, which indicated that they are in the same order. However, for the main topic of the study, I didn’t find the experimental results to support their results on the influence of compression and traction on the angiogenesis in the figure.
3. It seems that there are some mathematical symbols are misused, which leads to some confusion. For example, is the u_s in eq 8 same as u in eq 12 and 13? There is another u in eq 26. Are they have same meaning? Please check your writing carefully.
4. Many words in the figures are too small to be distinguished and read clearly.
5. The writing of the manuscript should be improved. Many sentences in the article are vague and not scientifically precise enough.
Reviewer 2 Report
Comments and Suggestions for Authors
The manuscript of the article «Computational Insights into the Interplay of Mechanical Forces in Angiogenesis» presented by Ana Guerra et.al. is of interest for studying the influence of compression and traction on angiogenesis, however, the article cannot be published in its presented form. It requires major revision.
First of all, the authors pay too much attention in the paper to the mathematical description of the Radial Point Interpolation Method. Moreover, this technical point is reflected in a large number of works cited by the authors (Belinha, J. Meshless Methods in Biomechanics - Bone Tissue Remodelling Analysi; and other papers).
At the same time, when the authors move on to modeling the angiogenesis process itself and the influence of stress on it, the completeness of the description disappears. So, for example, when describing 3 cases of different locations of VEGF generating regions, periodic boundary conditions apply ONLY for two of them. For the case of point localization of the VEGF producing region (middle picture in Figure 1), the specified periodic boundary conditions were not applied, which is clearly visible in the middle pictures of Figure 2.
The work does not indicate the elasticity parameters of the medium - the localized stress that the authors introduce in Figure 3 does not affect surrounding tissue, which is completely unrealistic.
When the authors describe the effect of pressure on VEGF diffusion, they point to the “alpha” parameter, which is also in Figure 4, but the parameter itself is not introduced anywhere!
The description of angiogenesis is also fragmentary and does not allow us to understand the results presented in Figure 6, where capillary growth is observed not along the VEGF gradient. Also the authors do not describe the capillary branching algorithm, but refer to their work (Guerra, A.; Belinha, J.; Mangir, N.; MacNeil, S.; Natal Jorge, R. Sprouting Angiogenesis: A Numerical Approach with 498 Experimental Validation. Annals of Biomedical Engineering 2020, 49, 871-884, doi:10.1007/s10439-020-02622-w.), where there is also no clear mathematical description of the algorithm for branching and stopping the growth of the vessel. Thus, it is not possible to accurately evaluate the results presented.
The paper needs to be completely revised, reducing the methodological part and significantly expanding the mathematical description of the angiogenesis process. It is also necessary to describe mathematically in detail the effects of external stress on tissue, VEGF diffusion and angiogenesis. All model parameters and boundary conditions used must be provided.
Round 2
Reviewer 1 Report
Comments and Suggestions for Authors
The authors have provided a detailed respone to my concerns. I think the edit is adequate, and I recommend publication.